# Surface Treatments of PEEK for Osseointegration to Bone

**DOI:** 10.3390/biom13030464

**Published:** 2023-03-02

**Authors:** Jay R. Dondani, Janaki Iyer, Simon D. Tran

**Affiliations:** McGill Craniofacial Tissue Engineering and Stem Cells Laboratory, Faculty of Dental Medicine and Oral Health Sciences, McGill University, 3640 University Street, Montreal, QC H3A 0C7, Canada

**Keywords:** dental implant, orthopedic implant, implant biomaterial, polymer, osseointegration, surface treatment

## Abstract

Polymers, in general, and Poly (Ether-Ether-Ketone) (PEEK) have emerged as potential alternatives to conventional osseous implant biomaterials. Due to its distinct advantages over metallic implants, PEEK has been gaining increasing attention as a prime candidate for orthopaedic and dental implants. However, PEEK has a highly hydrophobic and bioinert surface that attenuates the differentiation and proliferation of osteoblasts and leads to implant failure. Several improvements have been made to the osseointegration potential of PEEK, which can be classified into three main categories: (1) surface functionalization with bioactive agents by physical or chemical means; (2) incorporation of bioactive materials either as surface coatings or as composites; and (3) construction of three-dimensionally porous structures on its surfaces. The physical treatments, such as plasma treatments of various elements, accelerated neutron beams, or conventional techniques like sandblasting and laser or ultraviolet radiation, change the micro-geometry of the implant surface. The chemical treatments change the surface composition of PEEK and should be titrated at the time of exposure. The implant surface can be incorporated with a bioactive material that should be selected following the desired use, loading condition, and antimicrobial load around the implant. For optimal results, a combination of the methods above is utilized to compensate for the limitations of individual methods. This review summarizes these methods and their combinations for optimizing the surface of PEEK for utilization as an implanted biomaterial.

## 1. Introduction

The quest for materials that conform to the physiology of the human bone has been arduous in orthopaedics and implant dentistry. The human bone is a unique type of connective tissue as its form is determined by the nature of forces exerted on it. Therefore, a biomaterial used with bone has to mimic it mechanically and bind without inciting any local or systemic immune response. Orthopaedic and dental implants are medical devices that receive external load and transmit it to the bone. Dental as well as orthopaedic implants transmit stresses to the bone in a cyclic manner, making them susceptible to fatigue failure. Therefore, the implant biomaterial has to possess a similar modulus of elasticity (resistance to deformation) and fracture strength (stress level at which the material fractures) as that of human bone, while maintaining a robust interface with it. Materials like metals, autogenous bone, ceramics, polymers, and others have been used for orthopaedic implants. Metals, for example, have a high elastic modulus, which allows the metal to absorb stress that would otherwise be carried by the bone. This process is known as stress shielding and causes a reduction in bone density in vivo. The presence of metals also leads to the development of artifacts in radiographs. Autogenous bone grafts can cause bone defects at the donor site, leading to compromised form and function [1]. Ceramics have a similar limitation of high elastic modulus to metals, which is compounded by the inherent brittleness and lack of ductility. This can lead to an increased propensity for fracture of the prosthesis in vivo. Recently, polymers, in general, and PEEK have emerged as potential alternatives to conventional biomaterials. Due to its distinct advantages over metallic implants, PEEK has been gaining increasing attention as a prime candidate for orthopaedic and dental implants. PEEK has similar mechanical strength to bone. PEEK is radiolucent, chemically inert and resistant to sterilization, lacks metal allergies, and can be manufactured more easily [2,3,4]. PEEK has a modulus of elasticity similar to that of cortical bone, leading to proportionate flexure when loading is applied to the implant.

The phenomenon of osseointegration between the implant and bone warrants a hydrophilic surface for initiation by adsorption of plasma proteins (Figure 1). However, PEEK has a highly hydrophobic, bioinert surface that [a] attenuates osteoblast adhesion to PEEK surface, which prevents osteoblastic differentiation and proliferation, [b] leads to fibro-integration (foreign body reaction leading to a fibrous capsule around the implant instead of a bony union), and [c] weakens the contact between the implant and the bone (osseointegration), leading to implant failure, preventing clinical application [2,3,4,5,6]. Similar to other oral biomaterials, PEEK can accumulate oral biofilms that cause peri-implant or periodontal inflammation [7,8]. There are two types of inflammatory reactions around implants: reversible soft tissue inflammation known as mucositis and irreversible bone loss known as periimplantitis [9,10,11]. Since dental as well as orthopaedic implants osseointegrate with the bone for transmission of forces, the surface improvements of PEEK orthopaedic implants can also be translated to PEEK dental implants. It can be categorized into three main categories: surface functionalization with bioactive agents by physical or chemical means, incorporation of bioactive materials either as surface coatings or composites, and construction of three-dimensional porous structures on surfaces [12,13,14]. The physical treatments usually subject PEEK to plasma treatments of various elements, accelerated neutron beams, or conventional techniques like sandblasting and laser or ultraviolet radiation. The chemical treatments change the surface composition of PEEK and should be titrated against the time of exposure, and they can severely affect the performance of the bone–implant interface. In addition to these methods, the surface of the implants can be incorporated with bioactive materials [2]. The purpose of adding a bioactive material should be based on the desired use, loading condition, and antimicrobial load around the implant. This purpose dictates the extent to which the bioactive material will be incorporated with PEEK. First, the nanometer-scale coating can be applied to PEEK implants by spin-coating, gas plasma etching, electron beam deposition, or plasma ion immersion. Second, nanoparticles can be combined with PEEK through the process of melt-blending to produce bioactive nanocomposites [15]. The former is a surface-level phenomenon that does not affect the mechanical properties of the material, whereas the latter contributes significantly to the improvement of the strength and modulus of elasticity of PEEK. For optimal results, a combination of the aforementioned methods is utilized to compensate for the limitations of individual methods.

## 2. Background

Before considering PEEK as an implant biomaterial, a significant number of studies were conducted to assess and optimize the bonding strength of PEEK with other biomaterials. Parker et al. assessed different surface treatments and shear bond characteristics of PEEK to resin cement and glass ionomer cement. They concluded that sulfuric acid-treated samples had the highest shear bond strength [16]. However, it was evident that the physical alteration of the surface introduces a favourable morphology for adhesion but does not improve the hydrophilicity of the material. For the phenomenon of osseointegration to initiate between the implant and the bone, platelets and plasma proteins from the blood need to be adsorbed onto the implant surface. The surface of PEEK has to be altered to increase the surface energy and adsorption of adherends. Miyagaki et al. attempted the surface optimization of PEEK through the Friedel–Crafts reaction and demonstrated that the reaction followed by epoxidation increases the adhesion strength of PEEK to epoxy resins [17]. However, because of the direct modification of the main chains of PEEK, chemical treatments decrease its crystallinity, which must be preserved in order to maintain its mechanical properties [18]. It is essential to modify the depth of the surface in the direction of thickness in order to ensure sustained adhesion [19]. Lu et al. studied the effects of a plasma treatment on carbon fibers/PEEK hybrid composite and inferred that radiofrequency plasma activated by air, argon, or air–argon increases the interfacial strength of the composite. This is attributed to the increased roughness induced by the plasma treatment, leading to improved mechanical interlocking [20]. Among the various elements of plasma treatments, oxygen and hydrogen/oxygen plasma treatments were reported to be the most effective [21]. These studies have improved our understanding of the qualitative and quantitative reactions of PEEK to different treatments and how they can be employed in dental and orthopedic implantology.

## 3. Surface Treatments

Although a myriad of permutations and combinations of different surface treatments are employed to alter the surface topography of PEEK, for the sake of simplicity, these treatments have been classified into the following categories: physical treatment, chemical treatment, surface coating, and composite preparation (Figure 2), with the first surface treatment in the combination determining the classification. Though these terms are arbitrary and could lead to considerable overlap, physical and chemical treatments can be grouped into a subtractive form of surface modification while surface coating can be regarded as an additive form.

### 3.1. Physical Treatment

Physical treatments constitute plasma treatment, accelerated neutron atom beam (ANAB), photodynamic therapy, sandblasting, and laser irradiation.

#### 3.1.1. Plasma Treatment

Plasma treatments primarily aim to decrease the contact angle of the PEEK surface by increasing the surface energy. Secondarily, the plasma treatments incorporate the element constituting the plasma onto the surface of a PEEK (Plasma Immersion Ion Implantation). This improves its response to human osteoblasts. Several elements have been successfully tested for the plasma treatment of PEEK (Table 1).

As evident from the studies, plasma treatments are utilized as a pre-treatment for several other surface treatments. Oxygen plasma was tested with various forms of radiation, whereas Argon plasma was combined with various chemical treatments. However, the rationale for the preference for such combinations could not be deciphered. Considering the decrease in contact angle and increase in the bioactivity of PEEK, nitrogen plasma was found to be most suited for implant applications of PEEK. Plasma treatments and sulphonation are the most common surface treatments used to increase the surface energy of PEEK to receive a bioactive coating. The two treatments are also used extensively together. In addition to increasing the surface energy and hydrophilicity of the surface, plasma immersion ion implantation also increases the hardness of the surface [38]. This property may decrease the tribological wear of orthopaedic implants due to gliding surfaces but is not important for dental implants [39]. However, due to the nature of electromagnetic radiation, plasma treatments are restricted by line-of-sight limitations. Therefore, it is difficult to utilize plasma for modifying implants with complex geometry [40].

#### 3.1.2. Accelerated Neutral Atom Beam (ANAB)

Accelerated Neutral Atom Beam (ANAB) is a technique for employing an intensely directed beam of neutral gas atoms that improves the bioactivity of PEEK without altering its chemical or mechanical properties. Studies have demonstrated a decreased contact angle and increased bioactivity of osteogenic cells in response to ANAB-treated PEEK in-vitro [41,42,43,44] and an increased bond strength to bone in-vivo [42], as shown in Table 2. An in-vitro decrease in the bacterial colonization of Methicillin-resistant *Staphylococcus aureus* (*MRSA*), *Staphylococcus epidermidis* (*S. epidermidis*), and *Escherichia coli* (*E.coli*) have been demonstrated following treatment of PEEK with ANAB [43]. Though ANAB has exhibited significant improvement when used as a solitary treatment, its synergy with other treatments is yet to be tested. ANAB-treated PEEK surfaces are capable of osteoblast differentiation following osteoinduction in an osteogenic medium. However, ANAB-treated PEEK surfaces have not demonstrated independent osteoinduction ability [44].

#### 3.1.3. Photodynamic Treatment

Photodynamic treatment is primarily a therapeutic approach to decrease the microbial load on the surface. It involves the introduction of a drug on the surface of the biomaterial, followed by irradiation with a laser beam. It has been proven to decrease the microbial load and can be used in cases of inflammation around PEEK implants [45,46] (as given in Table 3). However, any potential role of photodynamic treatment in improving osseointegration in the absence of a periodontal pathology is yet to be investigated. However, significant improvements are required in light sources, absorption rates, and penetrating abilities of photosensitizers to decrease the exposure time required to modify PEEK surfaces with photodynamic treatment [46].

#### 3.1.4. Sandblasting

Sandblasting is a widely used procedure in the surface treatment of titanium implants. It was broadly used when the surface roughening of implants was first advocated to enhance osseointegration. It is a process of physically roughening the surface by subjecting it to a stream of abrasive particles. This method has been shown to increase the proliferation and differentiation of mesenchymal stem cells and also to mitigate inflammatory mediators around the implant [47] (Table 3). As observed in titanium implants, blasting is one of the most widely used surface treatments but is insufficient to improve tissue response and bone-implant contact. [38] Due to the advent of newer physical methods to treat PEEK surfaces, sandblasting has been restricted as a pre-treatment before the application of a bioactive coating.

#### 3.1.5. Laser

A femtosecond laser can be employed to induce periodic grooves on the surface of the PEEK implant, which improves the surface characteristics. Xie et al. confirmed that the PEEK surface after exposure to a femtosecond laser showed increased adhesion, proliferation, and differentiation of mouse bone marrow mesenchymal stem cells (mBMSC) cells and increased expression and activity of alkaline phosphatase [48] (Table 3). However, these findings will require substantiation with in vivo studies.

**Table 3 biomolecules-13-00464-t003:** Results of photodynamic therapy, sandblasting, and laser on PEEK.

Treatment	Result	Author
**Photodynamic therapy**		
(Temporfin/Ampicillin) + Diode laser	In vitro: Increase in resistance to microbial load	Peng et al. [46]
PDT/Sulphuric acid (H_2_SO_4_)/Air abrasion (Al/Diamond)	In vitro: Lower shear bond strength and microroughness of samples treated with PDT as compared to H_2_SO_4_ and Alumina particle air abrasion (Highest: H_2_SO_4_)	Binhasan et al. [45]
**Sandblasting**		
Alumina particles	In vitro: Increased proliferation and differentiation of rat MSCs and mitigation of inflammatory chemokine (C-C motif) Ligand 2 (CCL2)	Sunarso et al. [47]
**Laser**		
Femtosecond laser	In vitro: Increased adhesion, proliferation and differentiation of mBMSC cells and increased expression and activity of alkaline phosphatase	Xie et al. [48]

### 3.2. Chemical Treatment

#### 3.2.1. Sulphonation

One of the most common surface treatments employed for PEEK, sulphonation, involves the exposure of the surface to concentrated sulphuric acid. (H_2_SO_4_). The term ‘sulphonation’ refers to the exposure to H_2_SO_4_ as well as its removal from the surface by an alkali, although in some studies it has been used exclusively for the exposure to the acid. In Table 4, ‘sulphonation’ is inclusive of alkali treatment, or any other method employed to remove H_2_SO_4_ from the surface of PEEK.

Studies demonstrating the exposure time of H_2_SO_4_ have yielded conflicting results; however, most studies have employed an exposure time of 3–5 min. Ma et al. proved that an exposure time of 5 min to 98% sulphuric acid yielded optimal surface topography [49]. Cheng et al. demonstrated that a 3 min sulphonation decreased the contact angle and increased pre-osteoblastic activity [51]. On the other hand, Wang et al. found that a short exposure time of 30 s was optimal for decreasing the contact angle of PEEK; however, the longest time of exposure in the study was only 90 s [50]. If an additive coating will be applied after sulphonation, the time of exposure is determined by the other treatments used in the combination, as well as the nature of the coating. The time of exposure is a critical factor as the chemical composition of PEEK, which imparts superior mechanical properties to PEEK, gets altered as a function of time.

There is a statistically significant influence of sulphonation on the contact angle of PEEK. This phenomenon makes sulphonation acceptable as a pre-treatment before the application of a surface coating on PEEK. Furthermore, sulphonation has been extensively used in combination with plasma treatment for activation of the PEEK surface to receive organic and inorganic coatings and continues to be the most studied chemical treatment for the PEEK surface.

#### 3.2.2. Phosphonation

Phosphonation is the introduction of a phosphate group on the surface of a biomaterial. Various methods, such as diazonium chemistry and polymerization of Vinyl phosphonic acid, have been used to graft the functional group on the surface of PEEK, which has increased cell adhesion, spreading, proliferation, and differentiation in vitro and the bone–implant contact ratio in vivo [66,67,68], as shown in Table 5. It serves as an optimal surface treatment, but the compatibility of phosphonation with other treatments will have to be assessed.

#### 3.2.3. Silanization

Silanization is the introduction of a silane group to an object or surface. Silanization is utilized to improve the surface characteristics of materials like glass and metal oxides. It has been proven to increase the cell adhesion, spreading, proliferation, and differentiation of pre-osteoblasts in vitro (Table 5), but in vivo studies confirming the same are not available.

**Table 5 biomolecules-13-00464-t005:** Results of phosphonation and silanization on PEEK.

Treatment	Result	Author
**Phosphonation**		
Diazonium chemistry	In vitro: Decreased contact angle, increased deposition of HA and increased MC3T3-E1 cell viability and metabolic activity In vivo: Increased osseointegration	Mahjoubi et al. [68]
Vinylphosphonate	In vitro: Dose dependent increase in MC3T3-E1 cell metabolic activity In vivo: Dose dependent increase in bone-to-implant contact ratio and bond strength	Liu et al. [66]
Vinylphosphonate	In vitro: Increased MC3T3-E1 cell adhesion, spreading, proliferation and differentiation In vivo: Increased bone-to-implant contact ratio	Zheng et al. [67]
**Silanization**		
Dimethyl sulfoxide and Sodium borohydride + Silanization layers +Functionalization	In vitro: Increased MC3T3-E1 cell adhesion, spreading, proliferation and differentiation	Zheng et al. [69]

### 3.3. Surface Coatings

#### 3.3.1. Hydroxyapatite Coating

Hydroxyapatite (HA) is the main inorganic component of the human bone; hence, it is intuitive to incorporate HA on the surface of PEEK to increase its bioactivity. HA has regularly been employed as a surface treatment to increase the bioactivity of metallic implants. However, the temperature required to incorporate HA is higher than the temperature range at which PEEK is chemically stable. Hence, in most studies, an intermediate layer of Titanium or Yttria Stabilized Zirconia (YSZ) was used to shield PEEK from thermal insult [70,71,72] (Figure 3). The thickness of the intermediate layer has been found to influence the coating’s bond strength to PEEK. [72] HA coating is thermally treated to transform it into a crystalline state from an amorphous bioinert state. This finding is consistent with the fact that HA found in the human bone occurs in a crystalline state. In terms of bioactivity and bone strength to PEEK, heat-treated crystalline HA outperformed untreated amorphous HA [70,71,72,73] (Table 6).

#### 3.3.2. Titanium Coating

Titanium is the most widely used implant biomaterial due to its biocompatibility and osseointegration potential. The phenomenon of osseointegration was also accidentally discovered using titanium when titanium chambers that were used for studying the circulation of a healing fibula in a rabbit could not be removed due to their fusion with the bone [74]. A stable layer of titanium dioxide is formed on the surface of titanium on exposure to air [75,76]. This oxide layer has a high dielectric constant, which leads to the adsorption of proteins from blood on the surface, which is the first step in the cascade of events leading to osseointegration. Therefore, a coating of titanium or titanium oxide on the surface of PEEK would give the implant the favorable mechanical properties of PEEK and the higher bond strength of titanium with bone. In vitro studies have shown an increase in adhesion, proliferation, and differentiation of pre-osteoblasts in reaction to a titanium coating. In vivo studies have also inferred that the coating increases osseointegration and bond strength with bone (Table 7). However, titanium as an implant biomaterial has been reported to cause isolated cases of hypersensitivity in vivo [77], which currently remains unevaluated. Additionally, titanium and titanium alloys have a significantly higher elastic modulus than the supporting bone, leading to higher stress levels at the first point of contact on the bone. These stresses result in microfractures in the bone, leading to bone loss [78]. PEEK implants coated with titanium or titanium oxide benefit from the similar elastic modulus of PEEK and supporting bone, as well as the superior surface characteristics of titanium. Despite titanium allergies and mechanical mismatch with bone, titanium continues to be used in more than 92% of all dental implants and is a promising coating material for PEEK.

#### 3.3.3. Anti-Microbial Agent Coating

Anti-microbial coatings like gentamycin and selenium have been used in vitro [83,84] as well as in vivo [83] in combination with a carrier agent. These coatings have also been shown to increase the bioactivity of PEEK in addition to its antimicrobial activity against microbes, such as *S. aureus* and *Pseudomonas aeruginosa* (*P. aeruginosa*), as shown in Table 8. However, the sustainability of the release of these agents will have to be titrated against the complexity of the surface treatments to determine their feasibility. Additional studies quantifying antimicrobial action as a function of time are required to establish the durability of these coatings in an in vivo environment.

#### 3.3.4. Biomolecule Coating

Anti-inflammatory agents such as dexamethasone have been used to combat acute and chronic inflammatory responses of the body to noxious stimuli. Combinations of dexamethasone with other anti-inflammatory agents like interleukin-6 (IL-6) or metal-organic frameworks like Zn-Mg-MOF-74 have been proven to increase the anti-inflammatory response and antimicrobial activity of PEEK, respectively [86,87] (Table 9). Dexamethasone can be used as an implant coating in cases where the prognosis is compromised due to decreased local host immunity to infections, as supported by the current evidence.

#### 3.3.5. Polymer Coating

A coating of polymers like 2-methacryloyloxyethyl phosphorylcholine (MPC) on PEEK surfaces has been studied and shown to decrease the contact angle of PEEK, increase its wettability, and facilitate osseointegration [88] (Table 9). However, in vivo demonstration of the increase in hydrophilicity has not been documented.

**Table 9 biomolecules-13-00464-t009:** Results of biomolecule and polymer coatings on PEEK.

Treatment	Results	Author
**Surface coatings—Biomolecules**		
Dexamethasone + Nitrogen plasma treatment + IL-6	In vitro: Decreased peri-implant inflammatory mediators In vivo: Increased osseointegration	Xie et al. [86]
Zn−Mg-MOF-74 + Dexamethasone	In vitro: Increased antimicrobial activity against *S. aureus* and *E. coli* and angiogenic ability In vivo: Increased antimicrobial activity and angiogenic ability and osseointegration	Xiao et al. [87]
**Surface coatings: Polymers**		
2-methacryloyloxyethyl phosphorylcholine (MPC)	In vitro: Decrease in contact angle	Kyomoto et al. [88]

### 3.4. Composites of Poly (Ether-Ether-Ketone)

Composites of PEEK with various metals, oxides, inorganic fibers, and polymers have been used for improving the clinical performance of individual constituent biomaterials. The selection of material to be used with PEEK and the method of fabrication depend on the intended use of the implant. However, the main purpose of combining PEEK with other biomaterials is to improve the mechanical properties, with the improved surface characteristics being a by-product of the combination. Moreover, most composites of PEEK require an additional surface treatment for osseointegration. An exception in the following studies is a composite of HA and PEEK that did not require additional surface treatment and increased the adhesion, proliferation, and differentiation of pre-osteoblasts (Table 10), though in vivo studies are missing to substantiate the same. Most composites of PEEK are carbon-fiber reinforced PEEK (CFR PEEK) composites, which always require additional additive treatments for osseointegration.

## 4. Conclusions

PEEK is regarded as the future of dental and orthopedic implantology and is currently the only biomaterial that mimics bone biomechanically. However, its bio-inertness and hydrophobicity have restricted its commercialization as an implant biomaterial. PEEK’s low surface energy, which limits microbe colonization on its surface, also inhibits plasma protein adsorption and the adhesion and proliferation of undifferentiated mesenchymal cells on its surface, both of which are required for osseointegration.

Recently, numerous surface treatments that have been used conventionally to improve the implant surface characteristics of titanium have been applied to PEEK surfaces. The main objectives of these treatments are to increase the wettability and bioactivity of the surface as well as decrease the inflammatory mediators around it. These treatments either physically or chemically alter the surface or place a bioactive or antimicrobial coating on the surface. In practice, a combination of these methods is used to reduce the limitations of each individual method.

Among the physical treatments, plasma ion implantation is the most widely researched and used. Plasma treatments have demonstrated compatibility with nearly all other surface treatments, making them a preferred pre-treatment method. ANAB has also been proven to significantly improve surface characteristics, but it needs to be evaluated for compatibility with other treatments. Other surface treatments, such as photodynamic therapy, sandblasting, and femtosecond lasers, have been studied in a limited capacity, and their use as a surface treatment for PEEK will warrant additional scientific evidence.

Among the chemical treatments, sulphonation has universal acceptability and compatibility with other treatments. Sulphonation has been combined with most other surface treatments and has demonstrated exceptional synergism. The combination of plasma treatment and sulphonation has been employed as a pre-treatment for several bio-active coatings, further confirming that a combination of physical and chemical treatment is more effective than individual constituent treatments. Phosphonation has emerged as a promising treatment, but its effectiveness in a combination of treatments is yet to be evaluated.

Surface coatings have drastically improved the bioactivity of PEEK surfaces, although this effect is compounded by pre-treatment with physical or chemical treatments. HA and titanium coatings are the two most suitable treatments. HA application is particularly advantageous as it is the main constituent of the inorganic component of human bone. Crystallization of HA has been proven to improve the bioactivity of the PEEK surface considerably, and the intermediate layer used to thermally insulate the PEEK surface during crystallization does not decrease the bond strength of the coating. On the other hand, titanium is the most implanted material in human bone, which translates to adequate clinical use with academic evidence. In vitro and in vivo studies have substantiated the improvement in bioactivity and other surface characteristics of PEEK with the use of titanium as a coating after plasma treatment. Antimicrobial coatings have also been employed with success, but the inhibitory effects of these coatings over time will have to be studied further. Biomolecules and polymer coatings have shown favorable results in isolated studies, but additional evidence will be required to consolidate and confirm the results.

Composites of PEEK have also been attempted as a method to enhance the surface characteristics. Apart from composites of PEEK with HA, most other composites require an external surface treatment for their use as an implant biomaterial. Conventional classifications cite composites as a method to improve surface characteristics, but current studies have shown little evidence of the same.

Although these surface treatments have shown considerable in vitro and in vivo promise, there is a paucity of human studies confirming the same. In vivo studies cited in this work are conducted primarily on mice, rats, and rabbits and are short-term studies conducted over the past five years. The in vitro ‘efficacy’ and in vivo ‘effectiveness’ of these treatments, although demonstrated, mandate clinical ‘efficiency’. Some treatments will still require additional in vitro studies to establish repeatable and predictable results. Other treatments that have demonstrated in vitro efficacy require long-term human studies to justify their commercialization. Furthermore, currently, there are no studies comparing the clinical performance of commercially pure titanium with surface-treated PEEK to establish PEEK as a superior implant biomaterial after surface treatment. Nevertheless, the increase in the number of studies evaluating PEEK as an implant material is an attestation to the fact that PEEK is the future of dental and orthopedic implantology.

## Figures and Tables

**Figure 1 biomolecules-13-00464-f001:**
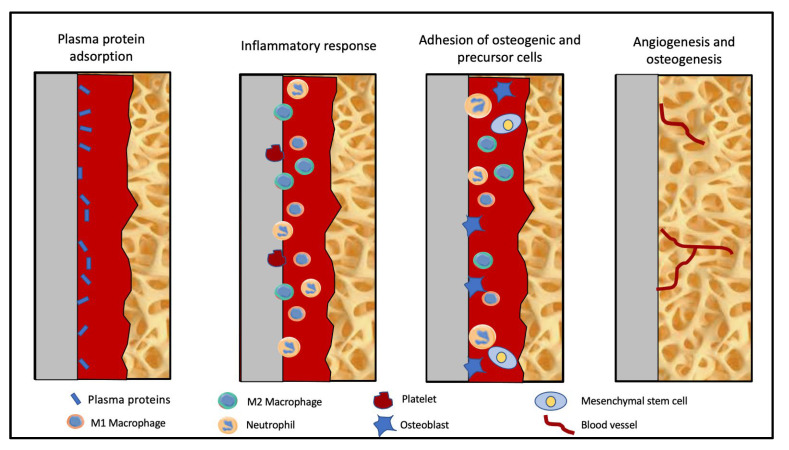
Process of osseointegration.

**Figure 2 biomolecules-13-00464-f002:**
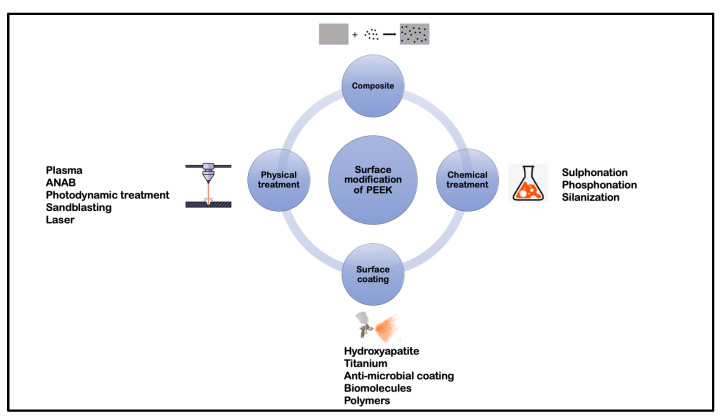
Classification of surface treatments of PEEK.

**Figure 3 biomolecules-13-00464-f003:**
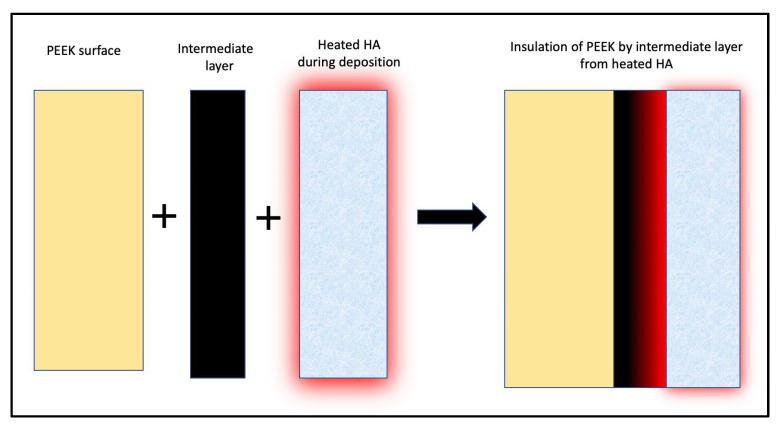
Insulation of PEEK by Titanium or YSZ intermediate layer from heated HA during crystallisation process.

**Table 1 biomolecules-13-00464-t001:** Various plasma surface treatments of PEEK.

Treatment	Results	Author
**Plasma**		
Oxygen/Ammonia	In-vitro: Increased adhesion, proliferation, and osteogenic differentiation of cells as compared to control	Althaus et al. [22]
Nitrogen	In-vitro: Increase in bioactivity and antibacterial properties with reference to *S. aureus*.	Gan et al. [23]
Oxygen/Argon	In-vitro: Increased wettability and cell adhesion, spreading, proliferation, and differentiation of SAOS-2 osteoblasts	Han et al. [24]
Oxygen/Nitrogen	In-vitro: Decrease in contact angle and no disadvantageous effect on cytocompatibility;	Ha et al. [25]
Nitrogen/Argon/(Nitrogen + Argon)	In-vitro: Increase in osteogenic activity (Highest: Nitrogen) and antibacterial property (Highest: Nitrogen + Argon)	Liu et al. [26]
Oxygen	In-vitro: Decrease in contact angle	Tsougeni et al. [27]
Oxygen	In-vitro: Increased cell adhesion and spreading of U2-OS osteoblasts in the presence of *S. epidermidis*	Rochford et al. [28]
Water vapour/Argon	In-vitro: Increased wettability and cell adhesion, spreading, proliferation, and differentiation of osteoblast precursor cell line derived from Mus musculus (mouse) calvaria (MC3T3-E1).	Wang et al. [12]
**Plasma treatment + Radiation**		
EUV + (low temperature Nitrogen/Oxygen)	In-vitro: Decreased contact angle and increased cell adhesion of MG63 cells, Cell adhesion higher with Nitrogen plasma	Czwartos et al. [29]
Oxygen/UV	In-vitro: Increase in the bond strength to TiO_2_ sol solution after exposure to O2 plasma/UV radiation	Kizuki et al. [30]
**Plasma + Chemical treatment**		
Argon + Hydrofluoric acid	In-vitro: Decreased contact angle and increased cell proliferation and differentiation of rBMS cells (Higher with Nitrogen) In-vivo: Increased resistance to *Porphyromonas gingivalis* (*P. gingivalis*)	Chen et al. [31]
Argon/(Argon + Hydrogen peroxide)	In-vitro: Increased cell adhesion, collagen secretion, and extra-cellular matrix deposition (Higher with Argon, Peroxide combination) In-vivo: Increased fibrous tissue filtration inhibition and osseointegration with Argon, Peroxide combination	Ouyang et al. [32]
**Plasma + Laser**		
Oxygen + Nd:YAG	In vitro: Decrease in contact angle	Akkan et al. [33]
**Plasma + Biomolecules/Inorganic coating**		
Argon + Polydopamine (PDA) + Vancomycin gelatin nanoparticles	In vitro: No cytotoxicity and increased antibacterial resistance to *Staphylococcus aureus* (*S. aureus*) and *Streptococcus mutans* (*S. mutans*)	Chen et al. [34]
Nitrogen + Tropoelastin	In vitro: Increased bioactivity of osteogenic cells	Wakelin et al. [35]
Nitrogen + PDA + Poly (lactic-co-glycolic acid) carrying Bone Morphogenic Protein-2 (BMP-2) gene	In vitro: Increased osteogenic activity	Qin et al. [36]
(Argon/Oxygen) + Acrylic acid vapours + Polystyrene sulfonate (PSS) and polyallylamine hydrochloride (PAH) multilayers	In vitro: Increased adhesion and proliferation of bone marrow stromal cells In vivo: Increased osseointegration	Liu et al. [37]

**Table 2 biomolecules-13-00464-t002:** Results of ANAB of PEEK.

Treatment	Result	Author
ANAB	In vitro: Decreased contact angle and increased bioactivity of osteogenic cells	Khoury et al. [41]
ANAB	In vitro: Increased wettability and cell adhesion, spreading, proliferation, and differentiation of SAOS-2 osteoblasts In vivo: Increased bond strength to bone	Khoury et al. [42]
ANAB	In vitro: Improved osteoblastic response and decrease in bacterial colonization of *MRSA*, *S. epidermidis*, and *E. coli*	Webster et al. [43]
ANAB	In vitro: Decreased contact angle and increased bioactivity of osteogenic cells	Ajami et al. [44]

**Table 4 biomolecules-13-00464-t004:** Results of sulphonation on PEEK.

Treatment	Result	Author
**Sulphonation**		
H_2_SO_4_+ (Acetone/Hydrothermal treatment/Sodium Hydroxide (NaOH) immersion)	In vitro: Optimal surface characteristics after 5 min exposure to 98% H_2_SO_4_; Comparable efficiency by Acetone, hydrothermal treatment and NaOH immersion in removal of residual acid	Ma et al. [49]
H_2_SO_4_+ NaOH	In vitro: Optimal contact angle reduction after exposure of 30 s to 98% H_2_SO_4_	Wang et al. [50]
H_2_SO_4_ + NaOH	In vitro: Decreased contact angle and increased bioactivity of MC3T3-E1 pre-osteoblasts cells	Cheng et al. [51]
**Sulphonation + Other chemical treatments**		
H_2_SO_4_/ [H_2_SO_4_ + Hydrogen peroxide (Piranha solution)]	In vitro: Decreased contact angle and increased adhesion and proliferation of human fibroblast cells	dos Santos et al. [52]
(H_2_SO_4_ + Nitric acid)/H_2_SO_4_	In vitro: Decreased contact angle and increased adhesion and proliferation of pre-osteoblasts cells (Highest with combination of H_2_SO_4_ and Nitric acid) In vivo: Increased bone formation around PEEK	Li et al. [53]
H_2_SO_4_ + Nitric acid	In vitro: Decreased contact angle and increased bioactivity of osteogenic cells	Huo et al. [54]
**Sulphonation + Organic/Inorganic coatings**		
H_2_SO_4_ + Lactams	In vitro: Decrease in growth on *S. mutans* biofilm	Montero et al. [55]
H_2_SO_4_ + zeolitic imidazolate framework-8 containing Ag ions	In vitro: Increase antimicrobial activity against *S. aureus* and, *E. coli*	Yang et al. [56]
H_2_SO_4_+ graphene oxide	In vitro: Increase in bioactivity and antibacterial activity against *S. mutans* and *P. gingivalis*	Guo et al. [57]
H_2_SO_4_+ [Simvastatin/Poly(lactic)acid] + Hyaluronic acid	In vitro: Increased MC3T3-E1 cell adhesion and proliferation	Deng et al. [58]
H_2_SO_4_ + Nickel nanoparticles + Hydroxyapatite	In vitro: Increase in angiogenesis and osteoblastic differentiation In vivo: Improved osseointegration	Dong et al. [59]
H_2_SO_4_+ lithium-ion-loaded Antimicrobial peptide (AMP)	In vitro: Increase in bioactivity and antibacterial activity In vivo: Increased antimicrobial activity and osseointegration	Li et al. [60]
**Sulphonation + Plasma + Coatings**		
H_2_SO_4_ + Oxygen plasma + alkaline Simulated Body Fluid (SBF)	In vitro: No cytotoxicity to MC3T3-E1 pre-osteoblasts In vivo: Increased osseointegration	Masomoto et al. [61]
H_2_SO_4_ + Oxygen plasma + Poly (Dopamine) + Tripeptide	In vitro: Decreased contact angle and increased bioactivity of osteogenic cells	Zhu et al. [62]
H_2_SO_4_ + Argon plasma + Polar amino functional groups	In vitro: Increase in bioactivity and antibacterial activity against *S. aureus* and *E. coli*	Wang et al. [63]
**Sulphonation + Other chemical treatments + Organic coatings**		
H_2_SO_4_ + Sodium borohydride + Phosphorylated gelatin + BMP-2	In vitro: Increased cell bioactivity of MC3T3-E1 pre-osteoblasts	Wu et al. [64]
H_2_SO_4_ + Nitric acid + Dopamine + Collagen	In vitro: Increased cell bioactivity of MC3T3-E1 pre-osteoblasts	Kim et al. [65]

**Table 6 biomolecules-13-00464-t006:** Results of Hydroxyapatite coating on PEEK.

Treatment	Result	Author
**Surface coatings—Hydroxyapatite**		
Hydroxyapatite	In vivo: Increased removal torque and biocompatibility	Johansson et al. [73]
[Hydroxyapatite/(Hydroxyapatite + Microwave annealing)] + YSZ intermediate layer	In vitro: Increased cell adhesion and proliferation with Hydroxyapatite crystallization with microwave annealing	Rabiei et al. [71]
Hydroxyapatite + Titanium intermediate layer + Hydrothermal treatment	In vitro: Bond strength of HA with PEEK with <10 nm Ti layer greater than that with >50 nm Ti layer	Ozeki et al. [72]
[Hydroxyapatite/(Hydroxyapatite + Microwave annealing + Autoclaving)] + YSZ intermediate layer	In vitro: Increased cell adhesion and proliferation with Hydroxyapatite crystallization with heat treatment	Durham et al. [70]

**Table 7 biomolecules-13-00464-t007:** Results of Titanium coating on PEEK.

Treatment	Result	Author
**Surface coatings: Titanium**		
Titanium [Pre-treated with grit blasting + Vacuum plasma (element unspecified)]	In vitro: Increased proliferation and differentiation of MC3T3-E1 cells In vivo: Increased osseointegration	Liu et al. 2021 [79]
Titanium + alkali treatment	In vitro: Increased adhesion and proliferation of pre-osteoblasts	Yang et al. [80]
(Oxygen plasma/Sandblasting) + Titanium sol + Hydrochloric acid	In vitro: Increased cell response In vivo: Increased osseointegration	Shimizu et al. [81]
Titanium dioxide (Pre-treatment: Argon ion + Titanium layer)	In vivo: Increased osseointegration and bond strength in pull-out test	Tsou et al. [82]

**Table 8 biomolecules-13-00464-t008:** Results of anti-microbial agent coatings on PEEK.

Treatment	Results	Author
**Surface coatings—Antibiotic agents with carrier**		
Brushite + Gentamycin sulphate	In vitro: Sustained biocompatibility and increased proliferation and differentiation of pre-osteoblastic cells In vivo: Increased antimicrobial resistance and osseointegration	Xue et al. [83]
Antimicrobial peptide (AMP) of GL13K/[AMP of GL13K + 1-ethyl-3-(3-dimethylaminopropyl) carbodiimide (EDC)]	In vitro: Increased antibacterial activity against *S. aureus*	Hu et al. [85]
Red selenium nanorods/Gray selenium nanoparticles	In vitro: Increased antimicrobial activity to *P. aeruginosa*	Wang et al. [84]

**Table 10 biomolecules-13-00464-t010:** Composites of PEEK and surface treatments.

Treatment	Results	Author
PEEK + Poly (ether imide) +Titanium dioxide coating	In vitro: Antibacterial resistance against gram-positive and gram-negative bacteria	Díez-Pascual et al. [89]
3D printed PEEK + crystalline Hydroxyapatite	In vitro: Increased adhesion, proliferation and differentiation of pre-osteoblasts and osteogenesis	Oladapo et al. [90]
Carbon reinforced PEEK + Zirconium ions using PIII	In vitro: Increased bioactivity of mBMSC cells and increased expression and activity of alkaline phosphatase, increased antibacterial activity against *S. aureus*, no effect against *E. coli*	Li et al. [91]
Carbon reinforced PEEK + H_2_SO_4_ + Oxygen plasma + Calcium phosphate	In vitro: Increased precipitation of apatite nuclei in SBF medium	Yamane et al. [92]
Carbon reinforced PEEK + H_2_SO_4_ + Dopamine HCl + Titanium carbide	In vitro: Evidence of photothermal antibacterial activity and cytocompatibility In vivo: Evidence of osseointegration	Du et al. [93]
Carbon reinforced PEEK + H_2_SO_4_ + Calcium chloride	In vitro: Increased precipitation of apatite nuclei in SBF	Miyasaki et al. [94]
Carbon reinforced PEEK + H_2_SO_4_ + Oxygen plasma + amorphous Calcium phosphate	In vitro: Increased precipitation of apatite nuclei in SBF medium	Yabutsuka et al. [95]
Carbon reinforced PEEK + H_2_SO_4_ + Hydroxyapatite	In vitro: Decrease in contact angle	Asante et al. [96]

## Data Availability

Not applicable.

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
