# Peer review of "Surface Treatments of PEEK for Osseointegration to Bone"

_biomolecules, 2023, doi:10.3390/biom13030464_

Round 1

Reviewer 1 Report

Review attached

Author Response

Reviewer 1:

In this manuscript, “Surface treatments of PEEK for osseointegration to bone.” the authors concluded some methods for optimization of the surface of PEEK as potential osseous implant biomaterials. The authors divided surface functionalization into three categories: (1) Surface functionalization with bioactive agents by physical or 16 chemical means, (2) incorporation of bioactive materials either as surface coatings or as composites, 17 and (3) construction of three-dimensionally porous structures on its surfaces. The review is detailed and comprehensive, and some problems need to be clarified before publishing in the Biomolecules.

Thank you for your comments. We have significantly improved the manuscript based on comments from both the reviewers, and hope this revised version will satisfy you, Thank you.

Please find our responses highlighted in yellow.

Following the text, a point-by-point list is given below.

Q1. The authors mentioned that PEEK has a highly hydrophobic, bioinert surface, which attenuates the differentiation and proliferation of osteoblasts. (Page 2, line 61) How can surface modification of PEEK change the property to promote osteoblasts adhesion?

The hydrophobic surface of PEEK prevents the adhesion of osteoblasts which hinders their subsequent proliferation and differentiation. By increasing the surface energy of PEEK by surface modification, osteoblastic adhesion is improved, that leads to the cascade of osteoblastic proliferation and differentiation. 

The manuscript has been modified as … However, PEEK has a highly hydrophobic, bioinert surface that [a] attenuates osteoblast adhesion to PEEK surface which prevents osteoblastic differentiation and proliferation, … (added on Page2, lines 99-101)

Kindly note that the references have changed in their numbers and order.

Q2. The authors mentioned PEEK as potential orthopaedical and dental implants. Though orthopaedical and dental implants are all implanted into craniofacial bones, there are still many differences in mechanical or biological requirements. Please add more data to discuss the main similarities and differences between orthopaedical and dental implants in the PEEK surface modification.

The similarities between orthopaedic and dental implants biomaterials and surface modification have been added to the revised manuscript as… Dental as well as orthopaedic implants transmit stresses to the bone in a cyclic manner making them susceptible to fatigue failure. Therefore, the implant biomaterial has to possess a similar modulus of elasticity (resistance to deformation) and fracture strength (stress level at which the material fractures) as the human bone, while maintaining a robust interface with it….(on Page 1 Lines 36-40) and

… . Since dental as well as orthopaedic implants osseointegrate with the bone for transmission of forces, these improvements can also be translated to PEEK dental implants….(Page 2 Lines 107-109)

Similarly, the differences between orthopaedic and dental implants surface modification have been added as … The two treatments are also used together extensively. In addition to increasing the surface energy and hydrophilicity of surface, plasma immersion ion implantation also increases the hardness of the surface.[39] This property may decrease the tribological wear of orthopaedic implants due to gliding surfaces but is not important for dental implants.[40] (on Page 6 Line 249-252.)

Kindly note that the references have changed in their numbers and order.

Q3. Many physical treatments like plasma, ANAB, photodynamic therapy, and sandblasting are introduced on Page 3. However, the shortcomings of each treatment should be considered.

Thank you for your comment and suggestion. The revised manuscript has been modified with the shortcomings of the aforementioned treatments, such as… However, due to the nature of electromagnetic radiation, plasma treatments are restricted by line-of-sight limitations. Therefore, it is difficult to utilize plasma for modifying implants with complex geometry.[41]… (added on Page 6 Line 252-254 - plasma treatment),

…. ANAB-treated PEEK surfaces have demonstrated the ability of osteoblast differentiation after osteoinduction in an osteogenic media. However, ANAB-treated PEEK  surfaces have not demonstrated the ability of osteoinduction independently.[45]… (Page 7 Line 297-299 ANAB),

… However, significant improvements are required in light sources and absorption rates and penetrating abilities of photosensitizer to decrease the exposure time to modify PEEK surface with photodynamic treatment.[47]… (Page 7  Line 310-312 - Photodynamic treatment), and

… ). As observed in titanium implants, blasting is one of the most widely used surface treatments but is insufficient to improve tissue response and bone-implant contact.[39] … (Page 8 Line 333-334 - Sandblasting)

Kindly note that the references have changed in their numbers and order.

Q4. As for surface coating, the authors illustrated HA coating, Titanium coating, biomolecule coating, polymer coating, and so on. Titanium implants are the most widely used dental implant worldwide, and what’s the superiority of PEEK implants with titanium coating, compared with titanium implants?

Thank you for this comment. We have modified the manuscript by discussing the superiority of PEEK implants with titanium coating when compared with titanium implants. The manuscript has been thus modified… Additionally, titanium and titanium alloys have a significantly higher elastic modulus than the supporting bone leading to higher stress levels at the first point of contact on the bone. These stresses result in microfractures in the bone leading to bone loss.[79] PEEK implants coated with titanium or titanium oxide benefit from similar elastic modulus of PEEK and supporting bone as well as superior surface characteristics of titanium. Despite titanium allergies and mechanical mismatch with bone, titanium continues to be used in more than 92% of all dental implants and is a promising coating material for PEEK… (Page 13 Line 460-465)

Kindly note that the references have changed in their numbers and order.

Q5. The review should discuss current shortcomings, future directions, and commercialization, along with barriers to clinical translation of these surface modifications of PEEK.

Thank you. We have revised the manuscript by adding some shortcomings, future directions and commercialization tips as discussed below,

… Although these surface treatments have shown considerable in-vitro and in-vivo promise, there is a paucity of human studies confirming the same. In-vivo studies cited in this work are conducted primarily on mice, rats, and rabbits and are short-term studies conducted over the past five years. The in-vitro ‘efficacy’ and in-vivo ‘effectiveness’ of these treatments, although demonstrated, mandate clinical ‘efficiency’. Some treatments will still require additional in-vitro studies to establish repeatable and predictable results. Other treatments which have demonstrated in-vitro efficacy require long-term human studies to justify their commercialization. Furthermore, currently there are no studies comparing the clinical performance of commercially pure titanium with surface treated PEEK to establish PEEK as a superior implant biomaterial after surface treatment….  

The current shortcomings are discussed on Page 18 Lines 642-646

Future directions are discussed on Page 18 Lines 646-648

Commercialization and barriers to clinical translation are discussed on Page 18 Lines 648-650

Kindly note that the references have changed in their numbers and order.

– We thank you Reviewer 1 for his/her comments and hope this revised version will satisfy you, Thank you.

Thank you.

Regards.

Reviewer 2 Report

A good review, but the manuscript suffers from numerous errors in English grammar, word use, punctuation, and sentence structure.

The tables are OK, but there are no figures. Most of us are graphic learners.

No consistency in the reference format. Some titles are in upper case, and others are in lower case.

Author Response

Reviewer 2

A good review, but the manuscript suffers from numerous errors in English grammar, word use, punctuation, and sentence structure.

Thank you for your comments. We have significantly improved grammar, word use, punctuation, and sentence structure of the manuscript based on comments from both the reviewers.

The tables are OK, but there are no figures. Most of us are graphic learners.

We have added figures to the manuscript (Figure no. 1, Page 2), (Figure no. 2, Page 4) and (Figure no. 3, Page 12).

No consistency in the reference format. Some titles are in upper case, and others are in lower case.

We have also modified the reference format.

We thank you Reviewer 2 for his/her comments and hope this revised version will satisfy you, Thank you.

Thank you.

Regards.

Round 2

Reviewer 2 Report

The authors have addressed my concerns.